# (Chemical) Roles of HOCl in Rheumatic Diseases

**DOI:** 10.3390/antiox13080921

**Published:** 2024-07-29

**Authors:** Jenny Leopold, Jürgen Schiller

**Affiliations:** Institute for Medical Physics and Biophysics, Medical Faculty, Leipzig University, 04103 Leipzig, Germany; juergen.schiller@medizin.uni-leipzig.de

**Keywords:** cartilage, synovial fluid, arthritis, ROS, myeloperoxidase, HOCl

## Abstract

Chronic rheumatic diseases such as rheumatoid arthritis (RA) are characterized by a dysregulated immune response and persistent inflammation. The large number of neutrophilic granulocytes in the synovial fluid (SF) from RA patients leads to elevated enzyme activities, for example, from myeloperoxidase (MPO) and elastase. Hypochlorous acid (HOCl), as the most important MPO-derived product, is a strong reactive oxygen species (ROS) and known to be involved in the processes of cartilage destruction (particularly regarding the glycosaminoglycans). This review will discuss open questions about the contribution of HOCl in RA in order to improve the understanding of oxidative tissue damaging. First, the (chemical) composition of articular cartilage and SF and the mechanisms of cartilage degradation will be discussed. Afterwards, the products released by neutrophils during inflammation will be summarized and their effects towards the individual, most abundant cartilage compounds (collagen, proteoglycans) and selected cellular components (lipids, DNA) discussed. New developments about neutrophil extracellular traps (NETs) and the use of antioxidants as drugs will be outlined, too. Finally, we will try to estimate the effects induced by these different agents and their contributions in RA.

## 1. Introduction

Rheumatic diseases such as osteoarthritis (OA) or rheumatoid arthritis (RA) are abundant diseases, especially in industrialized countries with high life expectancy of the citizens [1]. RA represents an autoimmune disease where the immune system attacks the joints and/or its constituents like proteins and/or carbohydrates, leading to persistent inflammation and damage. On the other hand, OA is a degenerative joint disease caused by excessive mechanical wear on joints, resulting in the cartilage destruction. The onset of RA is relatively rapid, while OA develops slowly over the years and regularly causes symptoms at elevated age. RA also affects organs and systems beyond the joints and may cause systemic symptoms such as fatigue, fever, and loss of appetite, which are typically not seen in OA. Although both diseases are symptomatically different, we will discuss them here simultaneously since both have the same unique (chemical) characteristics: the damaging and modification of the articular cartilage layer of the joint and/or the inflammation of the synovial membrane (synovitis) (Figure 1) [2].

In the USA alone, more than 40 million citizens (about 12% of the overall US population) suffer from arthritis [3]. The socio-economic consequences (including but not limited to the inability to work and early retirement as personal consequences) and the annual all-cost ranged from USD 12,509 to USD 20,919 per RA patient in 2019 [4]. Unfortunately, the incidence of the disease is also increasing. The reasons are unknown to date [3], but akinesia (meaning absent movement, i.e., the inability to perform a clinically perceivable movement [5]) as well as extreme sports may also be responsible for the observed increase. Although medial laymen often assume that the term arthritis represents just a single disease, it actually comprises more than 100 different types of arthritic diseases, which collectively affect virtually all parts of the human body—not only the joints [6].

Therefore, it is regrettable that neither a convincing cure of the disease nor reliable methods of early disease diagnosis are available—although there is currently intense research being performed [7]. Here, we will discuss changes in (articular) cartilage as well as the joint (synovial) fluid since both are affected by OA and RA, respectively.

## 2. Composition of Articular Cartilage and Synovial Fluid

Since both cartilage and synovial fluid (SF) have a rather unique composition (Figure 1), the architecture of (hyaline, articular) cartilage and the composition of SF will be shortly discussed. Further details are available, for instance, in [8,9]. With 70–80% of the wet tissue weight, water is the most abundant constituent of the cartilage [10]. This is crucial for maintaining the function of cartilage, i.e., its shock-absorbing properties and the frictionless motion of the bones within the joint [11]. So-called chondrocytes in the hyaline articular cartilage are responsible for both the synthesis and secretion of major components of the extracellular matrix (ECM) [12,13]. Cartilage ECM contains a particular network of highly hydrated proteoglycans and collagen fibrils [13].

### 2.1. Collagen

Different collagen types are expressed in articular cartilage (60–70% of the dry weight), which are essential for the elasticity and shear strength of the tissue caused by the presence of water between the collagen fibrils [14]. Collagen has a remarkable amino acid composition with high amounts of the amino acids glycine (Gly), proline (Pro) and hydroxyproline (Hyp) [15]. In particular, Hyp occurs exclusively in collagen, where it is essential for the noteworthy (thermal) stability of collagen [16]. Gly, Pro and Hyp mediate the triple-helical structure of collagen, leading to the formation of collagen fibers [8]. This makes (native) collagen insoluble in virtually all solvents. Only the denaturation of collagen into gelatin makes it soluble in hot water, which is widely used in the food industry [17]. Under physiological conditions, the insolubility of collagen is essential to prevent swelling (water-uptake) of the cartilage. Collagen degradation by proteolytic enzymes is therefore (in addition to further aspects) closely linked to cartilage degradation [18]. The in vitro degradation of cartilage by collagenase results in a strongly increased water content [19] under otherwise identical conditions. This has two different reasons: On the one hand, the water binding of the polysaccharides of cartilage is overestimated since the collagen moiety is partially lost. On the other hand, the swelling limitation of the collagen network is reduced. Thus, proteolytic enzymes were believed to represent the main effectors of cartilage degradation in the past [20]. Intact, triple-helical collagen cannot be digested by common proteolytic enzymes (such as trypsin), while denatured collagen (called gelatin; without collagen fibers) is highly sensitive to the majority of all proteases [19].

### 2.2. Proteoglycans

The main proteoglycan (aggrecan) in the hyaline cartilage consists of a core protein connected with both chondroitin and keratan sulfate chains [21], as shown in Figure 1. Chondroitin sulfate (CS) and keratan sulfate (KS), as well as hyaluronic acid (HA, as the backbone of the aggrecan aggregate), are known as glycosaminoglycans (GAGs) and are susceptible to degradation under oxidative conditions. Thus, oxidatively modified GAG species will be the focus of this review and their reaction behavior will be discussed in more detail compared to other cartilage constituents.

Accordingly, calcification processes of cartilage may lead to pathological conditions due to the reduction in the water content by shielding the negative charges of the native cartilage [22].

HA is the simplest GAG, just a polysaccharide made of glucuronic acid and N-acetylglucosamine disaccharide units, which are 1→3 glycosidically linked and lead to high-molecular-weight HA by 1→4 glycosidic linkages. HA is the only GAG without any sulfate residues and is only a minor constituent in the proteoglycan aggregates [23]. Nevertheless, HA is present in the cartilage as well as (in significant concentrations) in the SF. Its sulfated analogues such as KS and CS are much more abundant in the cartilage tissue. The sulfation of GAGs does not occur randomly but seems to be species-dependent. For instance, bovine cartilage has a higher content of the 4-sulfate compared to human cartilage [24]. There are also many indications that the ratio between sulfated and non-sulfated GAGs is altered in the aged cartilage [25]. Although this topic is outside the scope of this paper, there are also indications that the “sulfation code” plays a major role in signal transduction [26]. This indicates that the sulfation of cartilage polysaccharides does not occur randomly.

Intact SF is a highly viscous fluid and acts as joint lubricant and shock absorber, as well as an important source of nutrition for the articular cartilage, which borders on the synovial joint [27]. SF may be considered an “ultrafiltrate” of blood [28], i.e., the concentrations of salts and small molecules are nearly identical in both body fluids. Providing detailed compositional information about physiological SF is difficult since it can hardly be obtained from healthy volunteers due to ethical reasons. Thus, control samples are often missing in related studies. Furthermore, SF is a very viscous body fluid. This high viscosity is caused by the significant concentration of high-molecular-weight HA (about 2–3 mg/mL). Many attempts were performed to use the mechanical (viscoelastic) properties as a diagnostic criterion of degenerative joint diseases in the past [29]. This was motivated by the finding that the viscosity of SF can be reduced in the presence of reactive oxygen species (ROS) [30] released by stimulated neutrophilic granulocytes. Additionally, SF from patients with inflammatory joint diseases are characterized by significant myeloperoxidase (MPO) activities in the U/mL range (one unit (U) is defined as the amount of MPO that will reduce 1.0 µmol H_2_O_2_ per minute at 25 °C, pH 6.0) [31]. However, it has to be emphasized that the availability of SF to monitor pathological changes has nowadays significantly decreased: puncturing was still a very common medical tool a few decades ago since the majority of rheumatic patients suffered from swollen knees. This was reduced due to the development of more efficient drugs and treatments—particularly by “biologicals” [32].

## 3. Mechanisms of Cartilage Degradation

It is commonly accepted that inflammatory joint diseases affect both cartilage composition and layer thickness by the release of enzymes and/or ROS generation. However, there is still no consensus on the details about the mechanisms of cartilage degeneration [33]. Only two models are established:Chondrocytes (in the cartilage) or synoviocytes (at the interior of the articular capsule) are negatively affected by products of typical inflammatory cells such as neutrophilic granulocytes (vide infra). This leads to a reduced overall ECM synthesis and the increased degradation of the ECM [34]. Additionally, the “quality” of the de novo generated ECM may be poor (fibrocartilage instead of hyaline cartilage).Inflammatory cells such as neutrophilic granulocytes, macrophages or T cells release harmful enzymes that can degrade the ECM. Matrixmetalloproteinases (collagenase, for instance) are often assumed to be particularly responsible for that degradation processes [35], especially in combination with ROS. This opinion is emphasized by the observation that the number of granulocytes is augmented in the blood from RA patients [36] and that polymorphonuclear granulocytes (abbreviated as both PMNs or neutrophils) can be found in large numbers in the pannus tissue, a replacement tissue as a consequence of ECM degradation, of cartilage [37,38].

Although macrophages [39] and T-cells [40] may also contribute to cartilage degradation, this review will focus on products derived from neutrophils (most potent inflammatory cells), which are characterized by much higher concentrations of MPO than other cells [41,42]. The main reason why the contribution of neutrophils was underestimated to date is—at least partially—the fact that neutrophils cannot be kept in cell culture or only with considerable limitations (for example, as “neutrophil-like” cells [43]). Thus, they must be freshly prepared from the blood from volunteers [44] that confers many difficulties, particularly the poor reproducibility of the investigations. Recently, the “neutrophil extracellular traps” (NETs) were introduced as important constituents in RA [45] which will be discussed in Section 9 of this review (vide infra).

Normally, there is an equilibrium between the generation of ROS/reactive nitrogen species (RNS) and their consumption by antioxidants [46]. However, excessive ROS generation leads to oxidative induced post-translational modifications of proteins and may give rise to neoepitopes that are recognized by the immune system as non-self-substance. The resulting formation of autoantibodies is detectable by specific antigens, which might improve both early diagnosis and monitoring of disease activity. Particularly promising diagnostic autoantibodies include, for instance, anti-carbamylated proteins (generated by the reaction with a degradation product of urea) and anti-oxidized type II collagen antibodies [47].

## 4. Reactive Oxygen/Nitrogen/Chlorine Species

Such reactive species are generated in or (after the release of the required enzymes) from many cells [48]. For instance, fibroblasts, chondrocytes, macrophages, and, particularly, neutrophils are known as sources of ROS [49]. Neutrophils accumulate in inflamed joints in huge amounts [50,51] with an average percentage in the whole SF as high as 85.7% [52]. The physiological contribution of neutrophils (compared to the total cell number) is, however, much smaller.

Although pathogenic microorganisms are involved in some kinds of arthritis, antibodies, cytokines and chemotactic agents are assumed to represent one of the prime reasons for the accumulation and activation of neutrophils in the inflamed joint [36]. The increased oxygen consumption by neutrophilic granulocytes upon stimulation is called respiratory burst [53] and the generated ROS, RNS and reactive chlorine species (RCS) are summarized in Figure 2. These reactive species contain either oxygen, nitrogen, chlorine [54]—or even two of these (electronegative) elements. In some cases, the nomination is thus ambiguous: for instance, NO_2_ is commonly considered an RNS. However, it also contains oxygen. Therefore, ROS would be an appropriate term, too. We will use ROS throughout this review because it includes most of the relevant species—particularly HOCl.

ROS were comprehensively discussed already in the past [8]. Thus, we will only discuss those species, which are particularly relevant in the context of MPO and start with atmospheric oxygen (O_2_), which constitutes about 21% of the air. In a first reaction (Equation (1)), the enzyme NADPH oxidase catalyzes the reduction of O_2_ into superoxide anion radicals (O_2_^•−^), at which point the required electrons are generated by the oxidation of NADPH by the enzyme NADPH oxidase [55]:(1)$$NADPH+2O_{2}\overset{NADPH\;oxidase}{\to }{NADP}^{+}+H^{+}+2O_{2}^{•-}$$

Afterwards, O_2_^•−^ dismutates either spontaneously or particularly in the presence of the enzyme superoxide dismutase (SOD) into hydrogen peroxide (H_2_O_2_) and O_2_ (Equation (2)):(2)$$2O_{2}^{\;•-}+{2H}^{+}\;\;\overset{Superoxide\;dismutase}{\to }\;\;{H_{2}O}_{2}+O_{2}$$

H_2_O_2_ is the substrate of the enzyme MPO, which is particularly abundant in neutrophils, monocytes [56] and macrophages [57]. H_2_O_2_ is the educt of more efficient oxidants [58]. MPO reduces H_2_O_2_ to water under the formation of the so-called compound I (Figure 2). This activated form of the enzyme is reduced to the native enzyme either by abstracting two electrons from (pseudo)halides or by two one-electron steps via the formation of compound II [59]. In the first case, (pseudo)halides such as SCN^−^ are oxidized to (pseudo)hypohalous acids. The generation of hypochlorous acid (HOCl) and hypothiocyanate (OSCN^−^) is particularly important under physiological conditions [60]. The most important reaction can be summarized as follows:(3)$${H_{2}O}_{2}+{2Cl}^{-}\;\;\overset{Myeloperoxidase}{\to }\;\;HOCl+HCl$$

MPO is stored in the azurophilic (primary) granules within the neutrophils. The release of MPO may occur either into phagosomes containing engulfed pathogens or into the extracellular space, depending on the respective stimulus [61]. The most common degranulation involves the release of primary granules mainly into the phagosome, which presumably helps to prevent excessive tissue damage. However, in particular inflammatory conditions or in response to specific pathogens, larger moieties of MPO may be released extracellularly. This involves the fusion of granule membranes with the phagosome or the plasma membrane, enabling the release of granule contents into the extracellular space or phagosome [62], which is surprisingly accompanied by a significant change in the pH: immediately after phagosome formation, the pH rises to alkaline values due to NADPH oxidase activation. This pH change may facilitate MPO release and enhance MPO activity [63].

More recently, MPO degranulation was reported to be mediated via the extrusion of NETs [64], which are also discussed in this review (vide infra). It should be noted that controlled MPO release at the site of infection is of paramount importance for optimum activities. Any uncontrolled degranulation exaggerates the inflammation and may lead to unwanted tissue damage even in the absence of inflammation.

Furthermore, the determination of the MPO activity is one of the best diagnostic tools of oxidative stress biomarkers in arthritic diseases. Important activators and inhibitors of MPO are discussed in [61]: since high amounts of MPO (in the blood) are a decisive factor for early death [65], the development of suitable MPO inhibitors is of paramount relevance regarding the cure of arthritis.

Other halogenides such as Br^−^ are even more efficiently converted by MPO. However, hypobromous acid (HOBr) does regularly only play a minor role because it is by far less abundant than ${Cl}^{-}$ [66] at physiological conditions. Since HOCl is continuously generated by the $MPO/H_{2}O_{2}/{Cl}^{-}$ system, but immediately consumed by its reaction with (abundant) biomolecules, the assessment of the HOCl concentration under in vivo conditions is difficult. Accordingly, the in vivo HOCl concentration is presumably very low. However, some authors provided (significantly varying) HOCl concentrations. For instance, 50–100 mM were reported under inflammatory conditions [67], while 0.34 mM was reported as the HOCl concentration in the extracellular space [68]. The determination of the HOCl concentration in physiological systems is nowadays a hot topic [69] and there are many reports describing new methods of HOCl quantitation in physiological systems.

MPO is an abundant enzyme in neutrophils and constitutes about 5% of the entire protein mass within neutrophils [70]. Therefore, both MPO and HOCl as its prime product have been assumed to be massively involved in the degradation of the polymeric components (particularly GAGs of the proteoglycans) of cartilage for many years [68]. This aspect was confirmed by the fact that characteristic degradation products of cartilage (acetate and oligosaccharides) could be detected in the inflamed joint fluids from patients suffering from RA [31]. The contribution of neutrophils and products derived thereof in cartilage degradation has been recently reviewed [36]. Physicians recommend slight physical activity to their patients suffering from rheumatic diseases. In fact, physical activity may trigger the generation of MPO and, thus, the extent of HOCl generation [71].

## 5. Release of Enzymes into the Synovial Fluid—A Short Survey

The presence of MPO in SF was—to the best of our knowledge—first described in 1979 [72]. Hadler and coworkers determined the concentrations of several neutrophil-derived lysosomal proteins by immunochemical and enzymatic assays in 28 inflammatory and 9 non-inflammatory SF samples. The quantities of lactoferrin, MPO and enzymatically determined lysozyme correlated with the number of neutrophils. In contrast, enzymatic activities measured with synthetic substrates developed for the assay of chymotryptic-like cationic protein (cathepsin G) and elastase, along with immunochemically determined lysozyme, were independent of the number of neutrophils. The elastase activity (which was determined with elastin as substrate) was close to zero. A negative correlation between the concentration of common proteases and the degree of radiographic destruction of the joint could be observed. Cathepsin G and elastase are stored in an active form in neutrophil azurophilic granules [73]. There are two possible explanations why these proteins could not be detected: On the one hand, both enzymes may be missing and/or they are potentially not released by the primary granules of the neutrophils. On the other hand, inhibitors such as Thrombospondin 1 may considerably reduce the activities of these enzymes [74]. It seems likely that the elastase-alpha 1 proteinase inhibitor complex (EIC) plays a major role: for instance, the EIC levels increased according to the stage of articular cartilage destruction and the activity of neutrophil elastase was elevated in destructive joints of RA patients [75]. With the progression of articular cartilage destruction, the EIC levels in plasma of RA patients increased as well.

In a nutshell, the role of proteases is overstated in many studies in comparison to ROS. This can be understood by the used methods: proteases give defined reaction products, which can be rather easily determined, for instance, by mass spectrometry. Compared to that, the determination of ROS is much more difficult since the obtained products represent often transient products (e.g., chloramines).

## 6. Function and Modifications Induced by MPO

MPO plays a major role in the killing of invaded microorganisms by generating a particular ROS: compound I of MPO oxidizes chloride anions to HOCl, a strong oxidizing and chlorinating species (Figure 2) [76]. HOCl is a very weak acid with a pKa value of 7.53 [77]. At physiological conditions (pH = 7.4) there is, thus, a nearly 1:1 molar ratio between HOCl and the hypochlorite ion (ClO^−^). Although contributions of ClO^−^ cannot be ruled out at physiological conditions, there is nowadays an agreement that HOCl is the mainly relevant oxidizing species. Considering these aspects, one observation made in daily practice is somewhat strange: the product yield increases if the pH is lowered. Since HOCl should be reformed (according to Le Chatelier’s principle) from ClO^−^, a marked dependence on the pH could not be expected.

Dichlorine monoxide (Cl_2_O) may also be regarded as the anhydride of HOCl (Equation (4)) [78], which is increasingly assumed (in the same manner as chlorine gas [79]) to be involved in the deleterious actions of HOCl [78]. However, potential effects of Cl_2_O in the field of arthritis or cartilage degradation have not been investigated to date.
(4)$$2\;HOCl\;\to \;{Cl}_{2}O+H_{2}O$$

HOCl is an important source of other, more reactive ROS, such as hydroxyl radicals (${HO}^{•}$, Equation (5)) or singlet oxygen (O2 1, Equation (6)):(5)$$HOCl+{Fe}^{2+}\to \;{Fe}^{3+}+{HO}^{•}+{Cl}^{-}$$
(6)HOCl+H2O2→ O2 1+HCl+H2O

Converting nitrogen dioxide (NO_2_) into nitryl chloride (NO_2_Cl) by HOCl [80] may lead to serious modifications of the sidechains of proteins [81]. However, a detailed discussion of these aspects is beyond the scope of this review.

HOCl reacts with virtually all biomolecules, i.e., with amino acids, nucleic acids, sugars and lipids (Figure 3) [82] but with strongly different velocities. Comparative data of the corresponding reaction kinetics are available in different publications [83]. Thiols (as in cysteine) and thioethers (as in methionine) are the preferred targets of HOCl (indicated by the thick arrow in Figure 3).

After all thiol and thioether groups are consumed, other functional groups, such as amino functions in sugars, are affected by HOCl [84], i.e., HOCl undergoes well-defined, gradual reactions. The second order rate constants of the reactions between HOCl and thiols account for about 10^7^ M^−1^ s^−1^ [85] and with amino groups about 7 × 10^4^ M^−1^ s^−1^ [86], while the second order rate constants with olefinic residues are much slower and comprise only about 9 M^−1^ s^−1^ [87]. More detailed data were also compiled by Panasenko and coworkers [88] and by Davies et al. [89], and a survey of the data is available in Figure 4.

We will now discuss the reactions between HOCl as the main product of the enzyme MPO and the most important constituents of cartilage and SF, i.e., collagen, GAGs, DNA and lipids.

## 7. Reactions between HOCl and Cartilage Components

### 7.1. Collagen

The reaction between ROS and collagen was by far less comprehensively studied compared to the enzymatic degradation of collagen (see Section 2). Nevertheless, it is well known that the in vitro reactivity between selected amino acids and O_2_^•−^ or H_2_O_2_ (as the substrates of MPO) is poor [90]. Slight reactivity is exclusively observed at low pH values, at which particularly sulfur containing amino acids and (to a minor extent) aromatic amino acids are oxidized [90]. However, both are not very abundant in collagen.

Despite this poor reactivity, evidence has been provided that O_2_^•−^ producing systems contribute to tissue and cartilage degradation. At these conditions, solubilized collagen (isolated from skin or cartilage) can be assessed by an increased concentration of 4-hydroxyproline in the supernatant of the otherwise insoluble material. Similar results were also obtained upon incubation of cartilage specimens [91]. In addition to collagen degradation, GAG degradation (vide infra) was also monitored by an increase in the uronic acid concentration. Selective scavengers of O_2_^•−^ (SOD or catalase) decreased the extent of cartilage fragmentation. Despite these results, it can be assumed that O_2_^•−^ plays only a minor role in cartilage destruction due to its poor reactivity. It is much more likely that O_2_^•−^ is converted into a more deleterious species (particularly HO^•^) at pathological conditions. It is also reasonable to assume that iron ions are massively involved in these processes [92]. Nevertheless, there is a complex interplay between relevant enzymes such as SOD and ROS [93,94].

Although the reaction of HOCl and molecules like amino acids or carbohydrates is much faster compared to O_2_^•−^ or H_2_O_2_, there is no consensus about the HOCl-mediated effects on the collagen moiety of cartilage to date. On the one hand, it was shown that the collagen moiety in the cartilage is affected by HOCl to a lesser extent than the polysaccharides of articular cartilage [95]. Similar results were obtained when other types of cartilage, e.g., bovine nasal cartilage with an enhanced GAG content but a reduced collagen content, were treated with HOCl [96]. This is a clear indication that elevated concentrations of HOCl are required to affect the collagen moiety of cartilage. On the other hand, it became evident that HOCl activates collagenase, which may subsequently lead to damages of collagen [97]. This is presumably mediated by the modification of collagenase inhibitors, not the enzyme itself. In contrast, taurine chloramine (one of the very few stable chloramines) had the opposite effect and led to an inhibition of the collagenase activity [98]. Stamp and coworkers [99] found elevated protein carbonyl and 3-chlorotyrosine concentrations along with an increased MPO activity in the SF of RA patients. Since only MPO is able to generate 3-chlorotyrosine, the detection of this molecule is a strong hint on the production of HOCl. Of course, 3-chlorotyrosine is not necessarily derived from collagen but might also stem from the link or the core protein of the proteoglycans.

Odobasic and coworkers [100] explored the role of endogenous MPO in experimental (collagen-induced arthritis) RA in normal and MPO knockout mice. They found that MPO contributes to the development of arthritis: MPO enhanced the proliferation and decreased the apoptosis of synovial fibroblasts in vitro.

Westman and coworkers found that the chlorination of collagen type II (the most abundant collagen of cartilage [101]) might represent a mechanism by which immunoreactivity is induced and by which chronic joint inflammation is supported. However, the detailed characterization of the chlorinated collagen was not performed which is a significant weakness of this study [102]. Interestingly, physiologically relevant concentrations of HOCl (between about 5 and 50 μM) lead to the degradation of collagen [103] and reduce the gel-forming tendency of collagen. Similar effects are induced by chloramines (e.g., from amino acids), if they are used instead of the reagent HOCl. It was also suggested that hypochlorite, N-chloramines, and chlorine are involved in the oxidation of the pyridinoline cross-linkages within the collagen type II in articular cartilage during acute inflammation [104]. Somewhat later, the collagen type II modification by hypochlorite was investigated in more detail [105]. The authors found that chlorination decreases the radius of collagen II aggregates from 30 to 6.8 nm. Since this alteration already occurs at low concentrations of HOCl, changes in the aggregate size were suggested as the optimum markers of HOCl-induced collagen oxidation. In another study it was elucidated that the in vitro oxidation of collagen promotes the formation of advanced protein oxidation products and is involved in the activation process of human neutrophils [106]. Although cartilage-related collagen was less comprehensively studied, there is increasing evidence [66] that thyroid peroxidase and peroxidasin are key enzymes for thyroid hormone synthesis. This is also important regarding the establishment of functional cross-links in collagen IV during basement membrane development. Although less abundant in cartilage than collagen type II, the collagen IV network plays a crucial role regarding the mechanical integrity of the basement membrane. A key event represents unequivocally the formation of intra- and inter-collagen fibril crosslinks. An inter-residue sulfilimine bond, which does otherwise rarely occur, was reported to be unique by collagen IV. These crosslinks are particularly formed between lysine/hydroxylysine or methionine residues and might occur inter- and intrafibrillarly.

Due to its significance as the major crosslink in the collagen IV network, the sulfilimine bond plays a critical role in tissue development and various human diseases [107]. In that way, HOCl and particularly HOBr are very important [108]. The basic reactions are illustrated in Figure 5:

At sites of inflammation, such as the SF of RA patients, increased RNS-mediated protein damage has been detected in the form of a biomarker, 3-nitrotyrosine, by immunohistochemistry, Western blotting, ELISAs and MS. Further details are available in [109].

### 7.2. GAGs

There is a consensus that GAGs are more efficiently degraded by HOCl in the diseased joint than the collagen moiety [110]. This is caused by the poor solubility of collagen and its triple-helical structure, which aggravates the attack by ROS.

It has been established that the reagent HOCl (as well as the entire MPO/H_2_O_2_/Cl^−^ system) reduces the viscosity of solutions of high-molecular-weight hyaluronan. The term hyaluronan is a mixture of HA and hyaluronate which indicates that all species are included—independent of the charge state [111]. Baker and colleagues used size exclusion chromatography to evaluate the effects of either exogeneously added HOCl or the entire MPO/H_2_O_2_/Cl^−^ system on aqueous hyaluronan solutions. It could be shown that already µM concentrations of HOCl (HA was in the mg/mL range which corresponds—related to the molecular weight of the polymer repeat unit—to a mM concentration) reduced the viscosity of HA. In contrast, elevated concentrations of HOCl were necessary to reduce the molecular weight of the HA polysaccharide [111], i.e., to induce scissions of the glycosidic linkages between the individual monosaccharide units. This was explained by structural modifications of the HA polysaccharide occurring (in contrast to the cleavage of chemical linkages) already in the presence of moderate HOCl concentrations and is accompanied by the reduction in the viscosity of HA. A few years later an interesting phenomenon was reported [30]: HOCl/MPO depolymerizes only purified umbilical cord HA (in a HO^•^-dependent way) but does not depolymerize the HA polysaccharide in SF. The authors concluded that HOCl/MPO has a scavenging action on SF HA by the consumption of H_2_O_2_. In this way, the formation of HO^•^ radicals is suppressed. Therefore, aqueous HA solutions are not necessarily comparable with native SF since the aqueous solutions do not contain proteins [112].

The determination of the viscosity of (pathologically changed) SF was (and still is) an established method of the severity of inflammatory joint diseases after puncture of the affected joint [113,114]. Later studies emphasized the role of the proteins in biological materials—even if HA is still considered as a powerful mean to attenuate the oxidative stress in the inflamed joint: the injection of native HA into the joint is known to have several beneficial effects [114,115]. These studies were performed by monitoring potential effects of the inflammatory state by the measurement of some indicative effects such as the HA concentration and the protein composition on the SF viscosity [116].

The mechanism of HA degradation was studied by ^1^H nuclear magnetic resonance (NMR) spectroscopy in 1994. Both N-acetylglucosamine (GlcNAc) and chondroitinsulfate (CS) were separately treated with the reagent HOCl and the kinetics of the reaction monitored [117]. Since amino groups are particularly reactive, HOCl reacts first with the GlcNAc or the amino sugar within the GAG polysaccharide (Figure 6).

Two consecutive effects were observed: (a) the depolymerization of the GAG polysaccharide chain under scission of the glycosidic linkages; and (b) the cleavage of the N-acetyl groups via a transient, presumably chlorinated product. This product and the involved N-centered radicals could be later monitored by electron spin resonance and other methods [118]. Similar observations were made if isolated hyaluronan or HA from SF was subjected to γ-irradiation [119]: this leads to the scission of the water molecule under generation of HO^•^ radicals, which are capable of depolymerizing the HA molecule. Similar data may also be obtained if the radicals are generated via Fenton chemistry, which may also occur under in vivo conditions [120].

The potential contribution of N- and O-centered free radicals during the HOCl-induced degradation of HA is illustrated in Figure 7:

The relative reactivities of selected functional groups within the carbohydrates can also be determined by measuring the HOCl consumption: the required time-dependent data can be easily obtained by UV spectroscopy since the hypochlorite anion is characterized by an intense UV absorption (ε_290_ = 350 M^−1^cm^−1^). While different amino sugars possess significant reactivities, glucose or glucuronic acid do not react at all with HOCl [122].

Although there were no attempts to detect carbon dioxide (CO_2_) as a final oxidation product of HA and other carbohydrates to date, intermediate oxidation products such as formate are exclusively detected if a massive excess of HOCl is used. Since formate is an established product of HO^•^-induced carbohydrate degradation [120], this implies the conversion of HOCl into more reactive ROS (Equation (7)) [123].
(7)$${Fe}^{2+}+HOCl\;\to \;{Fe}^{3+}+{HO}^{•}+{Cl}^{-}$$

Akeel et al. [124] studied the in vitro reaction of HOCl with either HA or heparin (an agent against the consequences of arthritis [125]) using spectrophotometrically and enzymatically based methods. These authors found differences in dependence on the extent of sulfation of the involved carbohydrates. Although not yet completely clarified, the electron-withdrawing (-I) effect of the sulfate groups may play a significant role because it has an impact on the electron density at specific positions within the carbohydrate molecule.

It is very surprising that mass spectrometry (MS) has only been scarcely used to investigate the reaction between ROS and carbohydrates to date. Since oxidation is accompanied by changes in the molecular weight, MS represents a reliable method to monitor the ongoing changes. One of the available reports to date used electrospray ionization (ESI) MS to monitor the ROS-induced degradation of HA [126]. Unfortunately, this study gave only rather limited information but a clear indication that the investigation of oxidized carbohydrates by MS is even nowadays a challenging topic [127]. This is partially caused by the fact that sugars are rather refractive to the ionization process and give limited ion yields. This problem increases if the polarity of the carbohydrates increases, for instance by the introduction of sulfate groups [128].

In a more recent study, the chloramide of native, high-molecular-weight hyaluronan could be successfully synthesized. However, chlorinated isocyanuric acid had to be used as the chlorinating agent [129] because HOCl resulted in poor yields. In a very recent study, the degradation of HA and selected oligo- and monosaccharides by HOCl was studied by ESI MS in combination with thin-layer chromatography (TLC) [122]. It was demonstrated that the MS-based detection of N-chlorinated GAG amides is challenging while cleavages of the glycosidic linkages and the generation of chloramines in oligosaccharides are readily detectable. This study also provided evidence that the 1→4- and 1→3-glycosidic linkages exhibit different reactivities with HOCl [122].

In a previous study, pig articular cartilage was treated with different amounts of HOCl (at pH 7.4) and the composition of the supernatants assessed by ^1^H NMR [95]. Since collagen is very rich in glycine, the denaturation of collagen can be estimated by using the resonance at 3.55 ppm. However, this resonance was very weak and there were also no chlorinated products of glycine detectable [130]. In contrast, the intensities of GAG oligosaccharides (2.04 ppm) as well as acetate (1.90 ppm) were elevated subsequent to HOCl treatment of cartilage. This is a clear indication that the GAGs of cartilage are depolymerized by HOCl. Nevertheless, chlorinated peptides were also reported to be indicative of MPO activity [131]. These peptides are presumably not derived from the collagen but are stemming from other proteins, e.g., the core or the link protein of proteoglycans.

NMR is also a convenient method to discriminate SF from patients suffering either from OA or RA [132]. The particular advantage of NMR is that no prior knowledge about the sample composition is necessary.

Hawkins and Davies [133] made an interesting suggestion: MPO is a strongly cationic protein (positively charged, isoelectric point (pI) ≈ 10). In contrast, all GAGs within the cartilage are negatively charged, particularly the sulfated ones because sulfate represents a strong electrolyte. Due to the attraction of the differently charged molecules, it is likely that HOCl is generated in the vicinity of the GAGs. The short distance between MPO and/or the released HOCl may be one important reason why GAGs (similarly as negatively charged phospholipids [134]) represent one of the first targets of HOCl. This might explain the recent focus on NETs (vide infra).

### 7.3. DNA

Although it is well known that DNA reacts with HOCl [135,136], very little is known about this reaction in the context of RA and OA. Karaman and coworkers [137] reported DNA damage in RA lymphocytes in parallel with an increase in malondialdehyde (MDA) levels and decreased activities of SOD and glutathionperoxidase (GPx), which are both enzymes with antioxidative properties. These data emphasize again the increased oxidative stress in RA. A few years later, it was shown that polymorphisms of DNA damage repair genes play a role in RA pathogenesis. Accordingly, these DNA polymorphisms might be useful as RA disease markers [138]. Although a more detailed discussion of these aspects is beyond the scope of this review, it was very recently suggested that one hallmark of RA is impaired DNA repair observed in patient-derived peripheral blood mononuclear cells [139].

There are indications that products derived from NETs (vide infra) such as citrullinated histone H3 (H3Cit), cell-free DNA and MPO play a major role in arthritis and diagnosis [140]. These aspects are a promising research topic and were recently discussed in more detail [51,135].

### 7.4. Cellular Lipids

Since the cartilage tissue contains only comparably small amounts of lipids [141], the main source of lipids comes from the cells. Even if the cell density is rather poor (cartilage is known as a bradytrophic tissue) all cells contain many lipids (particularly phospholipids) in their cellular membranes and organelles [142]. In order to warrant a sufficient extent of membrane fluidity, the majority of membrane lipids contain double bonds [143]. Second order rate constant of HOCl against double bonds is rather poor. Nevertheless, lipids are affected due to their comparably high concentration in membranes [142].

As already indicated, amino-phospholipids such as phosphatidylethanolamine (PE) or -serine (PS) exhibit the most pronounced reactivity with HOCl [144]. However, there are no reports on chlorinated PE within the cartilage to date, although there is a very recent study on lipid markers between infrapatellar fat pad biopsies of OA and cartilage defect patients using MALDI MS Imaging. The main products between unsaturated lipids of the phosphatidylcholine (PC) type and HOCl are chlorohydrins, i.e., addition products of HOCl onto the double bond(s) [145]. Although this reaction has been well established for more than 30 years [146], it has to be emphasized that there are also free fatty acids due to the natural turnover of phospholipids in all biological systems. These free fatty acids react with HOCl under the formation of chlorohydrins as the main products as well [147]. There are indications that these products are converted into dimeric and trimeric products [148] with particular properties. Non-chlorinated (or in general non-halogenated) fatty acid esters of hydroxy fatty acids (FAHFA) are assumed to possess significant physiological relevance but are commonly generated by different pathways under the involvement of different enzymes [149]. Although it is well known that lipid chlorohydrins decrease the stability of membranes of erythrocytes [150], there is nothing known about the toxic effects of lipid chlorohydrins or chloramines on cartilage cells to date. However, in 2002, it was shown that 1,3-dichloro-2-propanol (1,3-DCP), an abundant impurity in many compounds like hard resins or celluloid, represents a major health concern [151].

Nitrite is known to be present in concentrations of up to 4 mM in SF from patients suffering from RA [152]. Therefore, it is likely that nitrite reacts with HOCl under generation of nitrate (Equation (8)).
(8)$${NO}_{2}^{-}+HOCl\;\to \;{NO}_{3}^{-}+HCl$$

Thus, the concentration of HOCl (vide supra) is reduced. This was first shown using the reaction of isolated lipids with HOCl [153]. Afterwards, it was also confirmed by systems mimicking the real conditions within the inflamed joint [154]: it could be convincingly shown that the presence of nitrite (but not nitrate) decreased HOCl-dependent cellular toxicity even if very small amounts of nitrite (in the µM range) were used. In contrast, nitrite (even in higher concentrations) did not inhibit superoxide-, hydroxyl radical-, hydrogen peroxide-, or peroxynitrite-mediated cytotoxicity. This is a clear indication of the different reactivities of the individual species. Additionally, the oxidation of plasmalogens by HOCl may also play a major role. These lipids are not characterized by two ester linkages (as it is the case in common lipids) but possess one ester and one alkenyl-ether linkage [155]. These lipids are relatively scarce in the SF [156] but give very characteristic oxidation products upon the reaction with HOCl since the resulting 2-chlorfatty aldehyde [157] has a high reactivity with many other functional groups, particularly amino residues.

A comprehensive review on the available “lipidomics” data on SF to date was recently published [158]. It was shown that PC and PE (the normally most abundant lipids) undergo major changes during the development of RA. A short summary of the most pronounced changes is shown in Figure 8:

## 8. Antioxidants to Suppress HOCl-Induced Effects

HOCl is a molecular (non-radical agent) with defined gradual reactivity: first, sulfhydryl groups are oxidized, and, afterwards, amino groups are converted into mono- or dichloroamines depending on the stoichiometry. The double bonds of lipids have the lowest reactivity and chlorohydrines are, thus, generated with the lowest velocity [133]. However, the reaction between HOCl and common antioxidants such as trolox (6-hydroxy-2,5,7,8-tetramethylchroman-2-carboxylic acid, an analogue of vitamin E) or ꞵ-carotene is faster compared to the majority of other biomolecules [89]. Compounds with reactive -SH or -S-S- groups such as lipoic acid [159,160] can be considered as effective antioxidants against HOCl although it is not yet completely clear whether the generated products are chemically inert and do not induce further reactions. Furthermore, different antioxidants were recently shown to be effective agents in the treatment of arthritis [161].

Taurine (2-aminoethanesulfonic acid) is often discussed [162] as the most important antioxidant against HOCl: the reaction between taurine and HOCl is slower compared to thiol groups but the products are characterized by significant stability [163]. This is important because there are also papers where the pharmacological aspects of the chloroamines are discussed. First, it was shown that the local administration of N-chlorotaurine represents an inhibitor of septic arthritis [164]. Additionally, N-chlorotaurine is discussed as a suitable topical (applied onto the skin) anti-infective. There is currently some research in order to develop novel mono- and dichloro- derivatives of dimethyltaurine, which are assumed to possess improved stability [165]. Although the pharmacological effects of well-known antirheumatic drugs such as methotrexate [166] and paracetamol [167] are surely different from the exclusive scavenging of HOCl, both are also known to react with HOCl. Therefore, one potential aspect might be the scavenging of HOCl.

The selective inhibition of MPO in the joint or the cartilage is unequivocally an effective way to suppress its effects and/or the generation of its most important product, HOCl. Compounds such as thioxanthines [168] or the KYC peptides (e.g., N-acetyl-lysyl-tyrosyl-cysteine amide) [169] seem to fulfill the corresponding needs. However, the long-term inhibitor use might be dangerous for the immune system. Many physicians are concerned about this problem and more research is needed in this field. The thioxanthines, for instance, have been shown to reduce, but not to prevent, the efficient killing of *Staphylococcus aureus* by neutrophils [170] and are, thus, potentially suitable for in vivo use. This aspect has been recently reviewed [171].

The fact that a knockdown of MPO is not lethal (the majority is clinically asymptomatic except if they are also diabetic [172]) may have two different reasons. Either residual MPO activity is sufficient for bactericidal activity or other enzymes can compensate for the inhibition of MPO: thioxanthines inhibit thyroid peroxidase and lactoperoxidase to a much lesser extent than MPO as assessed by in vitro assays [170]. In contrast, detailed studies with the KYC peptides are still lacking in the field of arthritis. Since chloride ions are substrates of MPO, reducing the intake of salt may have positive effects for arthritis patients: it was suggested that increased bone erosion under high-salt conditions can be attributed to an enhanced oxidative milieu maintained by infiltrating neutrophils [173].

Due to the incidence rate of RA and the limitations of reliable therapies, the search for new treatment strategies for RA became a transnational research focus. Specifically, ferroptosis is a novel type of programmed cell death characterized by iron-dependent lipid peroxidation. Ferroptosis is characterized by distinct differences from apoptosis, autophagy, and necrosis—and it may be triggered by ROS [174,175].

## 9. Neutrophil Extracellular Traps (NETs)

The fusion of both antimicrobial granules and the phagosome is an important event of neutrophils to kill invasive bacteria. A closely related new mechanism of neutrophils—called neutrophil extracellular traps (NETs)—was first described in 2004 [176]. The authors observed the release of different granule proteins and chromatin (simplified a mixture of DNA and basic proteins) by neutrophils upon in vivo activation by, e.g., lipopolysaccharides in order to attack pathological microorganism. Both compounds form extracellular fibers that are essential for the binding of bacteria and their subsequent killing.

Since this initial observation, the term and the discussion of NETs have been hot topics. NETs are defined as extracellular structures containing DNA, histones, and neutrophil-characteristic proteins such as MPO or elastase, which are secreted from the neutrophils in response to inflammatory stimuli. Additionally, neutrophils undergo chromatin remodeling with the subsequent release of the decondensed chromatin from the cells under generation of a network, illustrated in Figure 9.

Although many details of this mechanism are unknown to date [178], it is commonly accepted that NETs are produced in inflammatory areas and possess only a short half-life due to their fast enzymatic degradation [179]. Since a detailed discussion of all these events is clearly beyond the scope of this review, interested readers are advised to consult one of the more recent reviews [177]. Nevertheless, it is of interest to note that NADPH oxidase interactions with MPO are presumably involved which eventually lead to stimulation of neutrophil elastase. This is accompanied by the degradation of histones, chromatin decondensation and NET release.

This “suicidal NETosis” represents a new type of cell death in addition to apoptosis and necrosis [180] and with some peculiarities [181]. NETosis is nowadays assumed to represent a defense mechanism that is activated in response to the presence of inflammatory stimuli. In a nutshell, such processes are currently of huge interest regarding the development of antirheumatic drugs and the treatment of the disease in general [182]. Since our main interest is on the chemical aspects of cartilage degeneration, we will not delve into further details. However, a very crude survey of these processes is illustrated in Figure 10.

The role of NET detection in monitoring the severity of disease as well as the effect of different anti-rheumatic therapies is an important research topic. It seems that NETs play a major role in the pathogenesis of the disease and are not just a secondary finding associated with the inflammatory process. Further mechanistic processes (including immunological details) are available in [51,184]. It was also shown that neutrophils from RA SF drive inflammation through the production of chemokines, ROS, and NETs [51]. Inhibiting the formation of excessive NETosis, reducing the release of pro-inflammatory biological mediators, and converting the death form of neutrophils into a safer form of cell death represents presumably a key event. Finally, it is important to note that NETs are not only relevant in joint diseases but are also involved in important diseases such as atherosclerosis, sepsis and COVID-19 [185]. The available research literature in this field (between 1985 and 2023) has been recently reviewed [37].

## 10. Conclusions

Although clinically very different, RA and OA represent the most common musculoskeletal diseases, at which the synovia or the bone/cartilage is damaged at OA or RA conditions, respectively. One characteristic of OA is the degeneration of the articular cartilage, which is initiated by malfunctions of cartilage cells after joint injury. The degradation is additionally accelerated by inflammation, i.e., the generation of ROS and the release of tissue-degrading enzymes. The degeneration triggered by these biomechanical and biochemical mechanisms is irreversible. Although a lot of work was performed in the past, the complex mechanisms of cartilage degradation are insufficiently understood to date. One particular problem regarding the interpretation of the obtained data is the limitation of many studies on a particular compound class and/or a single degeneration-causing agent.

In particular, cartilage degeneration mechanisms on the cellular level have only been scarcely discussed to date, although there are indications that early post-traumatic biomechanical and inflammatory effects on cartilage cells impact the composition of cartilage [186]. Stimulated macrophages or neutrophils have only been scarcely used to mimic the “real” events of cartilage degradation to date. Since the events of cartilage degradation are very complex, there is an increasing number of in vitro models. For instance, it was suggested that the mitigation of post-injury inflammation leads presumably to the recovery of cartilage composition.

MPO is assumed to be massively involved in inflammatory joint diseases [99] for the following reasons:The number of neutrophils is massively enhanced in the SF from patients suffering from RA [187].Since MPO is a very abundant protein within the neutrophils (it constitutes about 5% of all proteins), the contribution of this enzyme and its product, HOCl, to cartilage degradation during RA is obvious.Characteristic chlorinated products (such as 3-chlorotyrosine [188] or HA oligosaccharides [31]) were found in the cartilage and/or the SF.

Thus, MPO seems to be an effective mediator in the development of rheumatic diseases, presumably accompanied by necrosis of the cartilage cells. Necrosis is also suggested to result in the release of damage-associated molecular patterns (DAMPs) and pro-inflammatory cytokines [189], leading to ECM degeneration caused by proteolytic enzymes [190]. Apoptosis, controlled cell death, has also been associated with excessive production of ROS [191]. Excessive ROS production has been suggested to promote ECM degeneration via decreased matrix biosynthesis [192]. However, further studies to improve the treatment of RA and OA are needed.

## Figures and Tables

**Figure 1 antioxidants-13-00921-f001:**
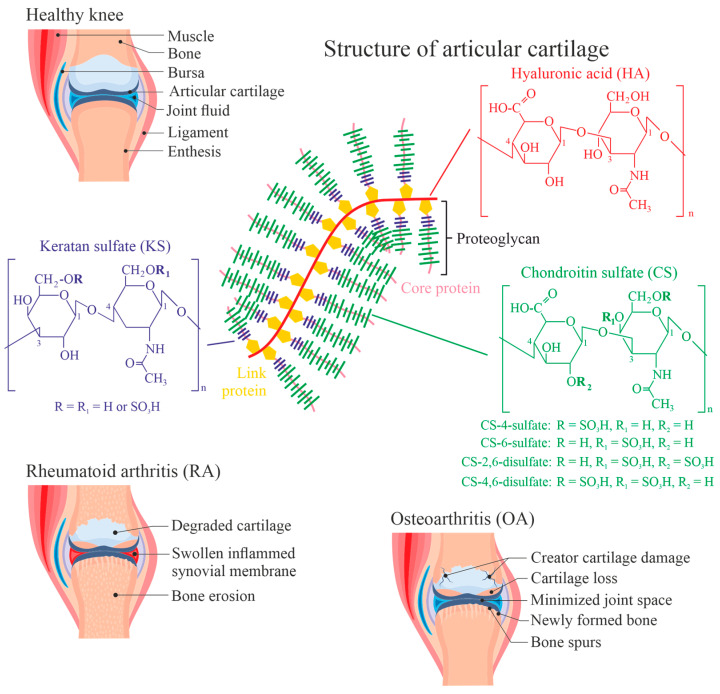
Survey of the most abundant glycosaminoglycans (GAGs) of healthy hyaline articular cartilage. As schematically shown, chrondroitin sulfate (CS, green colored) chains are more abundant than keratan sulfate (KS, purple colored) chains. Hyaluronic acid (HA, red colored), also named hyaluronan, functions as the backbone in the so-called “bottle-brush” structure. Structural changes in GAGs may lead to the degradation of the cartilage and thus to function loss of the cartilage. Characteristic patterns of healthy, rheumatoid arthritis (RA) as well as osteoarthritis (OA) cartilage are illustrated using stock photos from the illustration platform Colorbox (reproduced with permission using the campus license from Leipzig University).

**Figure 2 antioxidants-13-00921-f002:**
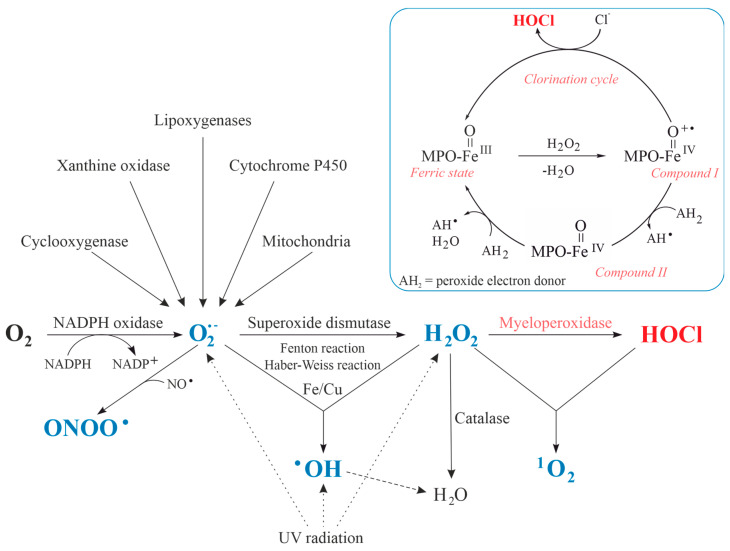
Pathways leading to and from different relevant reactive species (marked in blue). The HOCl generation is catalyzed by the enzyme myeloperoxidase (MPO) via the chlorination cycle (shown in detail, blue colored box). MPO does not exclusively metabolize Cl^−^, but Br^−^ and SCN^−^ as well.

**Figure 3 antioxidants-13-00921-f003:**
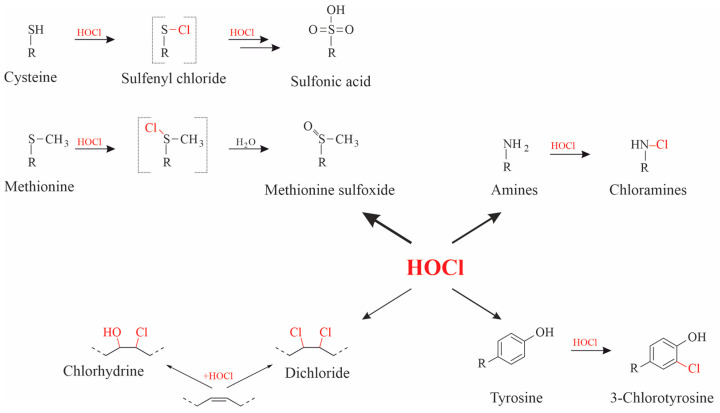
Overview about reaction products of different functional groups in biomolecules by HOCl. The thickness of the arrows is indicative of the corresponding reaction rates.

**Figure 4 antioxidants-13-00921-f004:**
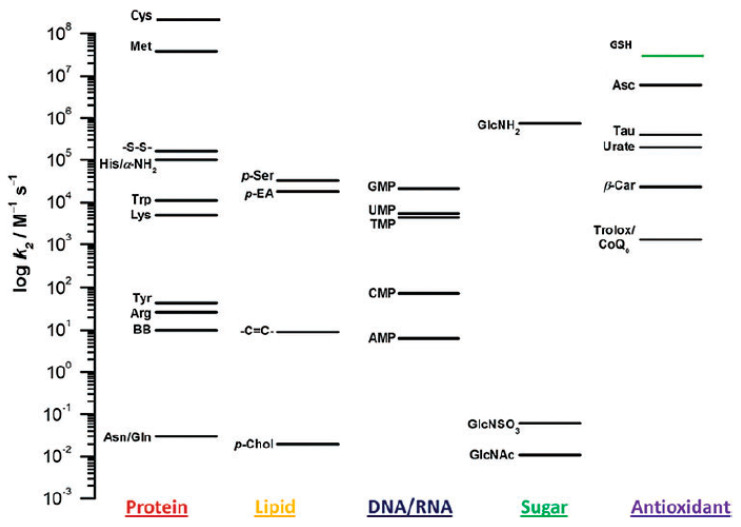
Plot summarizing the second-order rate constants (on a log scale) for the reactions of HOCl with model compounds of protein, lipid, and carbohydrate components, nucleobases, and antioxidants. Reprinted from [89] with permission from Mary Ann Liebert, Inc. (New Rochelle, NY, USA) The indicated k values may differ significantly in dependence on the method how they were determined. Therefore, relative reactivities are most relevant.

**Figure 5 antioxidants-13-00921-f005:**
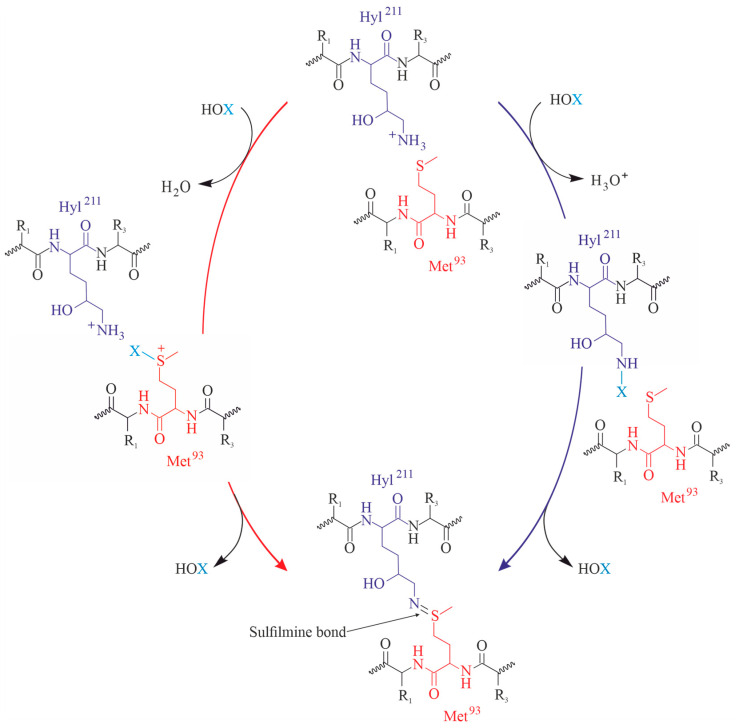
Proposed mechanism of sulfilimine bond formation in collagen IV referring to [107]. The oxidation is based on the reaction with hypohalous acid (HOX; X = Cl, Br) and the sulfur or nitrogen in methionine (purple) or lysine/hydroxylysine (red) residues, respectively. Both mechanisms lead to the generation of the sulfilmine bond.

**Figure 6 antioxidants-13-00921-f006:**
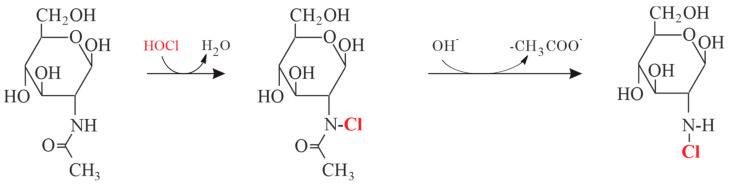
Pathway of the reaction between the reagent HOCl and N-acetylglucosamine as one important monosaccharide within the hyaluronan chain.

**Figure 7 antioxidants-13-00921-f007:**
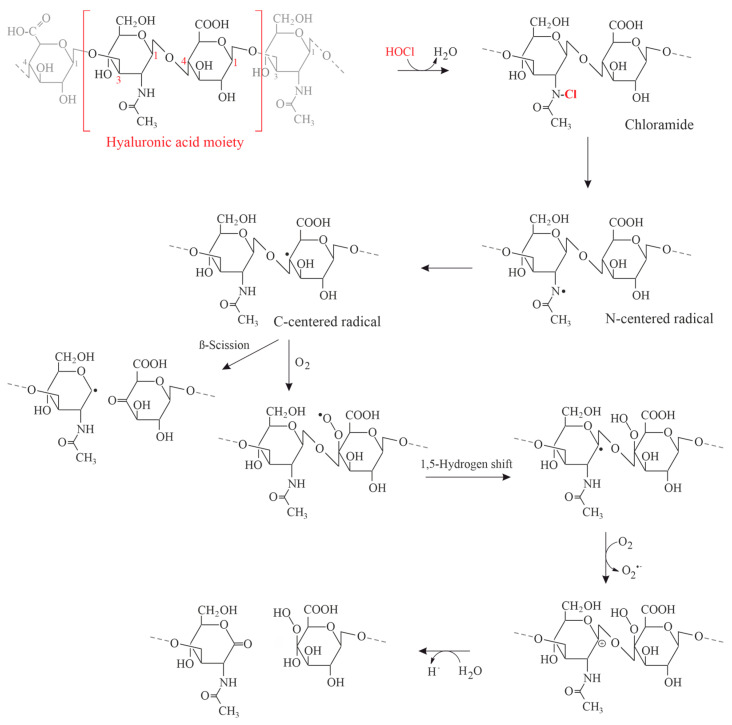
HOCl-induced fragmentation of the HA moiety as one representative example of GAGs. Reproduced from the work of Rees et al. [121] with permission and with slight modifications.

**Figure 8 antioxidants-13-00921-f008:**
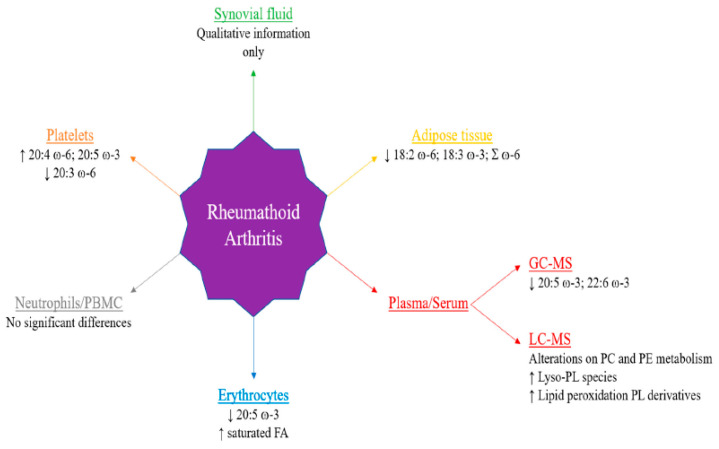
Main lipidomic alterations described in several studies of rheumatoid arthritis. Reprinted from [158] under the terms and conditions of the Creative Commons Attribution (CC BY) license.

**Figure 9 antioxidants-13-00921-f009:**
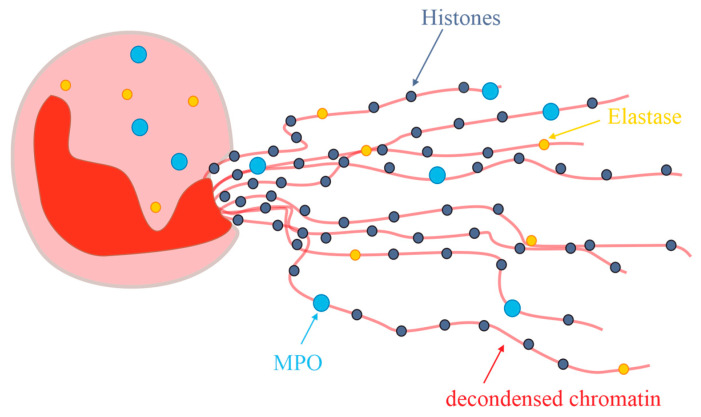
Schematic representation of Neutrophil Extracellular Traps (NETs) as network to kill invasive bacteria in accordance with [177]. Abbreviations: MPO, myeloperoxidase; NE, neutrophil elastase.

**Figure 10 antioxidants-13-00921-f010:**
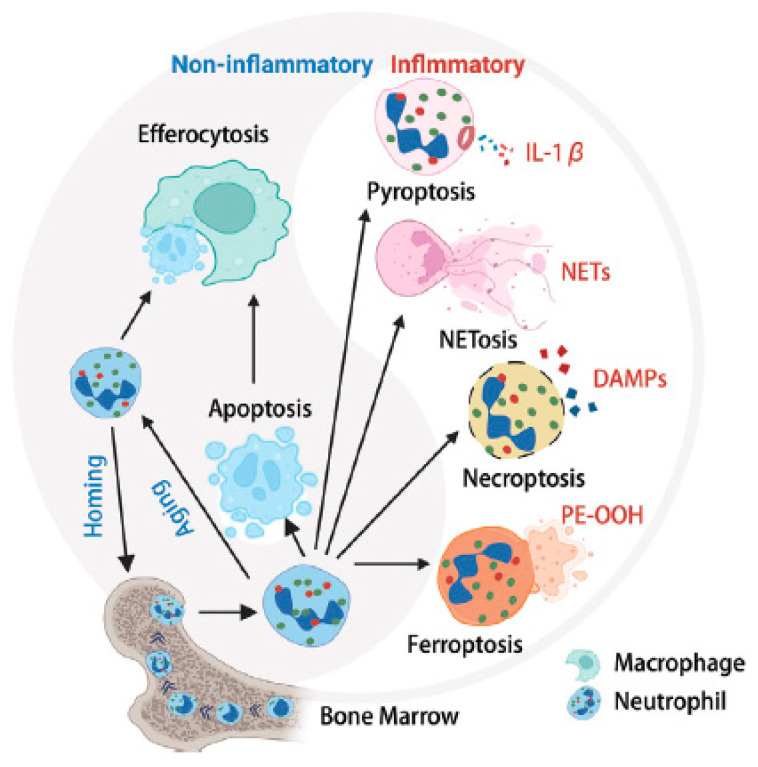
Neutrophil multifaceted death pathways in inflammatory conditions. Neutrophils are generated in the bone marrow through granulopoiesis and subsequently enter the circulatory system. Depending on the specific microenvironment, neutrophils undergo various mechanisms of cell death. These mechanisms encompass both non-lytic apoptosis and lytic death modalities, including necroptosis, pyroptosis, ferroptosis, and NETosis. These lytic processes are accompanied by the release of cytotoxic cellular proteases, cell-free DNA, and chromatin into the microenvironment. Each death pathway operates through distinct molecular mechanisms and regulatory networks, ultimately resulting in either immunosuppressive or pro-inflammatory outcomes. Defects in the clearance of apoptotic neutrophils and the accumulation of cellular remnants contribute to the onset of inflammatory diseases and autoimmune disorders (Reprinted from [181] under the terms and conditions of the Creative Commons Attribution (CC BY) license). Abbreviations: DAMP, Damage-Associated Molecular Patterns; IL1-ꞵ, Interleukin 1-ꞵ; NET, Neutrophil Extracellular Traps; PE-OOH, Phosphatidylethanolamine Hydroperoxide (The detection of anti-phosphatidylethanolamine autoantibodies will not be discussed here) [183].

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
