# Peer review of "(Chemical) Roles of HOCl in Rheumatic Diseases"

_antioxidants, 2024, doi:10.3390/antiox13080921_

Round 1

Reviewer 1 Report

This manuscript aimed to review the potential role of HOCl in rheumatic diseases. I have the following comments for the authors` consideration.

1. The manuscript focus appears to be shifting at places from RA through osteoarthritis to inflammatory arthritis. The Abstract mentions discussion of the roles of chlorine gas and dichlorine monoxide in inflammation, but these cannot be located within the text.  

2. The text is disproportional as well-established mechanisms are described in detail, whereas the description of recent developments (e.g. pyroptosis or DNA) is rather terse.   

3. The authors propose that HOCL represents a common mechanism to evoke cartilage damage in RA and osteoarthritis. However, these pathologies exhibit distinct clinical and macroscopic features. How could these be reconciled with a common mechanism?

4. What is the link between neutrophil MPO and DNA damage in RA lymphocytes?

5. What is the link between polymorphism in DNA damage-repair genes and oxidative stress?

6. What evidence indicates the beneficial effects of MPO inhibition in the cartilage? In which form of arthritis? How would this be “dangerous for the immune system”?

7. Neutrophils are thought to contribute to tissue damage through releasing of extracellular traps, which could occur in ROS-dependent manner. This issue warrants discussion.

8. Another important omission is that neutrophils are increasingly being recognized to possess pro-healing properties.

9. The focus on neutrophils should be better explained. For example, why were macrophages ignored?

The manuscript will benefit from extensive editing. Few examples:

Line 39. “Considering the USA as one selected country” should read “In the USA alone…”

Line 136. Units are arbitrarily defined, hence reference to U/ml range may not be informative.

Line 149. “Inflammation cells” should read “inflammatory cells”.

Line 154. What are the “common inflammation cells”?

Line 159. Reference to neutrophils should be uniform throughout the text (e.g. PMNs or neutrophils).

Line 164. PMN is a standard abbreviation for neutrophils.

Line 165. What does “most potent inflammation cells” refer to?

Line 166. The statement is incorrect. While neutrophils are short-lived cells, they can be kept in culture for a limited time. The argument does not make sense.

Line 176. What is “the normal contribution of neutrophils”?

Line 178. Chemotactic agents are fairly well characterized.

Line 185. Nitrogen-containing radicals are commonly referred to as RNS (reactive nitrogen species) and not ROS.

Line 205. MPO is also expressed by a subset of macrophages. Would this cell type be relevant to arthritis?

Line 246. The statement does not make sense. In what context was the role of proteases overstated?

Line 319. Statement on reactivity should be refined.

Line 377. “it is established since many years” should read “it has been established”.

Line 396. The comparison of HA aqueous solution and native SF is unclear.

Line 398. In other place in the text the authors argue against the use of biopsy in rheumatoid diseases. Which one is correct?

Line 468. “In an elder study” should read “In a previous study”.

Line 488. What does “first targets” mean?

Line 518. “There are increasing indications” should be rephrased.

Line 566. “topical anti-infective” of what?

Line 579. It is insufficient to refer to another review.

Line 581. Partial MPO deficiency is fairly common in the Caucasian population and this does not seem to increase the risk of infections.

Line 586 should be revised.

Line 588. In addition to ferroptosis and necrosis, necroptosis may be triggered by ROS. However, cell death in various types of cells have different consequences. For instance, apoptosis in inflammatory cells may facilitate the resolution of inflammation. These warrant more detailed discussion

Line 590. It is insufficient to refer to another review.

Line 613. In what type(s) of arthritis?

Line 615. The statement should be revised.

Line 623. What is a “subacute form of cell death”?  

Author Response

Reviewer 1:

This manuscript aimed to review the potential role of HOCl in rheumatic diseases. I have the following comments for the authors` consideration.

Thank you for the careful reading of our manuscript and your - overall - appreciation of our work. We have tried to deal carefully with your comments and hope that you will be satisfied with our revised manuscript version and the changes we have made.

  1. The manuscript focus appears to be shifting at places from RA through osteoarthritis to inflammatory arthritis. The Abstract mentions discussion of the roles of chlorine gas and dichlorine monoxide in inflammation, but these cannot be located within the text.

We feel sorry for this. Actually, our viewpoint is that from chemists (not a physician), i.e. we consider diseases ending with "itis" as inflammatory diseases accompanied by the generation of ROS. In our revised manuscript, we pay somewhat more attention on the differentiation of the different diseases. The aspects related to Cl2 and Cl2O are now emphasized in our revised manuscript in more detail (the discussion of these aspects in our original manuscript at page 7 was somewhat scarce) but both, Cl2 and Cl2O are not mentioned anymore in the abstract. Thank you again for this comment.

  1. The text is disproportional as well-established mechanisms are described in detail, whereas the description of recent developments (e.g. pyroptosis or DNA) is rather terse.

We feel sorry for this. We have written the review according to our expertise, which is on chemical aspects of rheumatic diseases. This is also the reason why we have chosen "(Chemical) Roles of HOCl in rheumatic Diseases" as title of our work. Both indicated aspects are now explicitly mentioned in our revised manuscript and new chapter entitled "Neutrophil Extracellular Traps (NETs)" was introduced. However, the related discussion is short and a more detailed discussion of these aspects is below the scope of our review. There are reviews available, which focus exclusively on these aspects. Some adequate reviews are cited in the new chapter.

  1. The authors propose that HOCl represents a common mechanism to evoke cartilage damage in RA and osteoarthritis. However, these pathologies exhibit distinct clinical and macroscopic features. How could these be reconciled with a common mechanism?

We have clarified this aspect and explained that both diseases are regarded (from our "chemical" viewpoint) as inflammatory diseases, which are both characterized by the infiltration of neutrophils which can be monitored by characteristic products within the synovial fluid. Thank you for bringing this point to our attention.

  1. What is the link between neutrophil MPO and DNA damage in RA lymphocytes?

Thank you for this important (and very good question). This aspect would surely merit its own review and can only be loosely discussed in our review with the focus on chemical aspects. Therefore, we kept the corresponding paragraph rather short since a comprehensive treatise would go beyond the scope of our review. This particularly applies because we are no experts in this field. Hopefully, you are satisfied with our approach.

  1. What is the link between polymorphism in DNA damage-repair genes and oxidative stress?

A detailed discussion of this point would be useful. However, we are no experts in this field (cf. our comments above) and have no major knowledeg about this topic. Therefore, only minor amendments were made in our manuscript.

  1. What evidence indicates the beneficial effects of MPO inhibition in the cartilage? In which form of arthritis? How would this be “dangerous for the immune system”?

There is some evidence that HOCl plays a major role in cartilage destruction. This particularly applies because MPO is a strongly positively charged protein. Since the polysaccharides of cartilage are negatively charged, it is likely that they will "bind" to MPO. Thus, ROS are generated in the vicinity of the cartilage polysaccharides and these will be particularly affected. It is our assumption that smaller amounts of HOCl (i.e. reduced MPO activities) induce smaller damages of cartilage. We have tried our best to make this clearer for the readers of our review.

  1. Neutrophils are thought to contribute to tissue damage through releasing of extracellular traps, which could occur in ROS-dependent manner. This issue warrants discussion.

This aspect is now discussed in much more detail and a dedicated chapter dealing with these aspects has been added to our manuscript. This was also requested by the second reviewer. Thank you for pointing out this important aspect. We hope that you will be satisfied with this additional chapter.

  1. Another important omission is that neutrophils are increasingly being recognized to possess pro-healing properties.

We respectfully disagree with the reviewer. There is so far only a handful of papers on this particular subject. Therefore, only minor changes were made and this aspect is just mentioned.

  1. The focus on neutrophils should be better explained. For example, why were macrophages ignored?

We respectfully disagree with the reviewer - macrophages are not ignored but are explicitly mentioned in the manuscript. Since they occur in smaller numbers in the synovial fluids from patients with rheumatic diseases and possess smaller amounts of MPO, neutrophils are more intensely discussed than amcrophages.

The manuscript will benefit from extensive editing. Few examples:

Line 39. “Considering the USA as one selected country” should read “In the USA alone…”

Changed as suggested.

Line 136. Units are arbitrarily defined, hence reference to U/ml range may not be informative.

We respectfully disagree with the reviewer: according to our best knowledge enzyme activities (in contrast to the amounts of enzymes) are commonly given as "Units". Anyway, the meaning of MPO "activity" is now defined and this aspect discussed in more detail.

Line 149. “Inflammation cells” should read “inflammatory cells”.

Changed as suggested.

Line 154. What are the “common inflammation cells”?

Thank you for your careful reading. Examples of such cells (neutrophils, macrophages, T-cells, etc.) are now given.

Line 159. Reference to neutrophils should be uniform throughout the text (e.g. PMNs or neutrophils).

"Neutrophils" is now used throughout the text.

Line 164. PMN is a standard abbreviation for neutrophils.

We agree. This abbreviation is now explained in our manuscript.

Line 165. What does “most potent inflammation cells” refer to?

Since neutrophils are the most abundant cells in synovial fluids from arthritis patients and have the largest amounts of MPO we consider them as the "most potent inflammation cells".

Line 166. The statement is incorrect. While neutrophils are short-lived cells, they can be kept in culture for a limited time. The argument does not make sense.

Is this really true? To our best knowledge neutrophils cannot be kept in culture (or only for very short periods) but have to be isolated from blood or synovial fluid from volunteers. Do you maybe mean "neutrophil-like cells" (Yokoyama et al. Cells 12 (2023) 322)? Anyway, this point is now also discussed in our review and an additional reference provided which supports our viewpoint [Hallberg et al.: Antioxidants (Basel) 12 (2023) 478].

Line 176. What is “the normal contribution of neutrophils”?

What we meant is the number of neutrophils in "healthy" synovial fluids. Problems in the determination of this parameter are now explained. In a nutshell, the availability of synovial fluid from healthy donors is very limited due to ethical limitations.

Line 178. Chemotactic agents are fairly well characterized.

This is a true comment. "(widely) unknown" was deleted in this sentence.

Line 185. Nitrogen-containing radicals are commonly referred to as RNS (reactive nitrogen species) and not ROS.

This is true. We have tried our best to clarify this problem in our revised manuscript version.

Line 205. MPO is also expressed by a subset of macrophages. Would this cell type be relevant to arthritis?

This is indeed true. Due to the large number of neutrophils we have focused, however, on these cells. Nevertheless, the potential contribution of macrophages is now also mentioned and shortly discussed.

Line 246. The statement does not make sense. In what context was the role of proteases overstated?

The activities of selected proteases such as elastase can be rather easily determined. In contrast, the "concentration" of ROS is much more difficult to determine since they are short-lived species. Therefore, many scientists focused on the proteases. This is now explained in more detail in our manuscript.

Line 319. Statement on reactivity should be refined.

This is now more carefully explained. The reactivity of HOCl (and other ROS) with the collagen and the glycosaminoglycans was meant.

Line 377. “it is established since many years” should read “it has been established”.

Changed as suggested.

Line 396. The comparison of HA aqueous solution and native SF is unclear.

We feel sorry that this statement was not clear enough: HA is a main constituent of SF and solutions of HA (with a high molecular weight) mimic the viscosity of synovial fluid. This is now explained in more detail.

Line 398. In other place in the text the authors argue against the use of biopsy in rheumatoid diseases. Which one is correct?

Obtaining healthy material is very difficult due to ethical reasons (puncture of healthy joints?). This is now also shortly explained.

Line 468. “In an elder study” should read “In a previous study”.

Changed as suggested.

Line 488. What does “first targets” mean?

This is now explained in more detail. HOCl reacts with different functional groups but with different velocities (or more exactly with different second order rate constants). This was meant by this statement.

Line 518. “There are increasing indications” should be rephrased.

Thank you. This was rewritten.

Line 566. “topical anti-infective” of what?

This is now explained. Effects of drugs applied as wound dressings are meant.

Line 579. It is insufficient to refer to another review.

Why? This review contains a number of useful references.

Line 581. Partial MPO deficiency is fairly common in the Caucasian population and this does not seem to increase the risk of infections.

This aspect is now mentioned and MPO-knockout and MPO deficiency are now strictly differentiated.

Line 586 should be revised.

We changed this line and hope the meaning is now easier to comprehend.

Line 588. In addition to ferroptosis and necrosis, necroptosis may be triggered by ROS. However, cell death in various types of cells have different consequences. For instance, apoptosis in inflammatory cells may facilitate the resolution of inflammation. These warrant more detailed discussion.

These aspects are now discussed in more detail and a dedicated chapter on "Neutrophil Extracellular Traps" was introduced.

Line 590. It is insufficient to refer to another review.

Why? This review contains a number of useful references and is, thus, more useful than giving some selected references to selected references. Nevertheless, an additional, original paper was introduced.

Line 613. In what type(s) of arthritis?

Explained now. It is rheumatoid arthritis.

Line 615. The statement should be revised.

What do you see wrong with this statement? We do not have problems with it and left this statement unchanged.

Line 623. What is a “subacute form of cell death”?

Sorry for this funny statement. "subacute form" was deleted.

Overall, thank you again for the careful reading and the helpful comments.

Reviewer 2 Report

The revised version of this manuscript is well-written and provides an interesting overview of the field of HOCl and RA. I have a few additional comments:

Comments:

-        A short section about neutrophil degranulation and MPO release is needed. Is MPO active upon degranulation? Is it released in granules, exosomes, or as a free enzyme? Are there any inhibitors present?

-        The authors state that cathepsin G and elastase activity are not relevant or are close to zero in SF samples. Please explain the underlying mechanisms. Are there endogenous inhibitors present? Are the enzymes inactive upon neutrophil degradation?

-        The MPO KO mouse data should be included (PMID: 24757143).

Minor comments:

-        The US has approximately 333 million citizens; please adjust the statistic regarding 10% of arthritis patients.

-        Line 246: one ‘also’ needs to be deleted.

see above

Author Response

The revised version of this manuscript is well-written and provides an interesting overview of the field of HOCl and RA. I have a few additional comments:

Thank you for the careful reading and your overall appreciation of our manuscript. We have tried our very best to deal carefully with your comments and addressed all your critique points.

Comments:

-      A short section about neutrophil degranulation and MPO release is needed. Is MPO active upon degranulation? Is it released in granules, exosomes, or as a free enzyme? Are there any inhibitors present?

Thank you for this excellent question. In our revised manuscript version, the aspects you have mentioned are now discussed at appropriate positions within our revised manuscript. We would be happy if you would be satisfied with the changes we have made.

-      The authors state that cathepsin G and elastase activity are not relevant or are close to zero in SF samples. Please explain the underlying mechanisms. Are there endogenous inhibitors present? Are the enzymes inactive upon neutrophil degradation?

This is an excellent question. Thank you again for the careful reading. Honestly, we are not absolutely sure about this points and some aspects remain speculative. Nevertheless, this point is now discussed in a bit more detail in our revised manuscript.

-      The MPO KO mouse data should be included (PMID: 24757143).

This paper is now explicitly cited and the messages from this paper are shortly summarized in our revised manuscript. Thank you for pointing out this important aspect.

Minor comments:

-    The US has approximately 333 million citizens; please adjust the statistic regarding 10% of arthritis patients.

Thank you again for the careful reading. This sentence has been adjusted.

-           Line 246: one ‘also’ needs to be deleted.

Changed as suggested.

Reviewer 3 Report

The present review covers the role of MPO-mediated damage in RA and OA , pathologies with apparently different etiology.

The authors should focus in a more detail whether MPO-mediated damage and thus in particular HOCl is the only driving factor in chronic and/or inflammatory conditions or if other mechanisms or even cell types beyond the neutrophils (not representing the only source of MPO) may act as additional driving factors promoting the disease (please see line 163). Maybe the role of HOCl-modified lipids (MPO-mediated plasmalogen species) in addition to HOCl-modifed proteins OA and RA could be worked out in a more detail. Furthermore, a potential role of reactive nitrogen species generated via the MPO-H2O2 system could be addressed in the progression of OA and RA.

Another suggestion would be whether NETs are (in)directly involved in RA and OA and whether HOCl and/or MPO-mediated DNA damage might play a proper role as promoting species as reported in atherosclerosis and other diseases. Furthermore, it might be of interest whether neutrophils and their “different subspecies” might only exert adverse effects or even beneficial ones that would counteract with an enhanced and adverse activity of the MPO-H2O2-Cl-/Br- system in OA and RA. The authors further refer to a potential involvement of chlorine gas (Cl2) and dichlorine monoxide (Cl2O) in the abstract, however, this issue has been superficially addressed (lines 259-260) and thus does not justify any comment within the abstract.

please see above: major comments

Author Response

The present review covers the role of MPO-mediated damage in RA and OA, pathologies with apparently different etiology.

Thank you for the summary of our manuscript. Your critical, but helpful comments are really appreciated and we have revised our manuscript accordingly. Hopefully, you will be satisfied with this result.

The authors should focus in a more detail whether MPO-mediated damage and thus in particular HOCl is the only driving factor in chronic and/or inflammatory conditions or if other mechanisms or even cell types beyond the neutrophils (not representing the only source of MPO) may act as additional driving factors promoting the disease (please see line 163).

The contribution of cells other than neutrophils is now explicitly mentioned. However, the focus of this special issue of "Antioxidants" is on MPO (and similar enzymes) which are present to the highest extent in neutrophils and macrophages. Therefore, our focus remains on neutrophils, which are also the most abundant cells in the pathologically-changed synovial fluids.

Maybe the role of HOCl-modified lipids (MPO-mediated plasmalogen species) in addition to HOCl-modified proteins OA and RA could be worked out in a more detail.

Thank you for this important comment. The role of oxidatively-modified lipids is now discussed in more detail. The reaction products of plasmalogens and HOCl (such as 2-chlorofatty aldehyde) are now also discussed - Although the content of plasmalogens in the synovial fluid is rather poor.

Furthermore, a potential role of reactive nitrogen species generated via the MPO-H2O2 system could be addressed in the progression of OA and RA.

Reactive nitrogen species are surely of interest - and this is already reflected by the comparably high nitrite content of synovial fluids. As far as we can say, however, a comprehensive review of the impact of reactive nitrogen species would exceed the size of a readable review. Therefore, we have not included a comprehensive discussion but discuss this aspect just shortly.

Another suggestion would be whether NETs are (in)directly involved in RA and OA and whether HOCl and/or MPO-mediated DNA damage might play a proper role as promoting species as reported in atherosclerosis and other diseases.

"Neutrophil Extracellular Traps" (NETs) are now discussed in much more detail and a dedicated chapter dealing with these aspects has been added to our manuscript. This was also requested by the first reviewer. Thank you for pointing this important aspect out. We would be happy if you would be satisfied with the way we have dealt with your comment.

Furthermore, it might be of interest whether neutrophils and their “different subspecies” might only exert adverse effects or even beneficial ones that would counteract with an enhanced and adverse activity of the MPO-H2O2-Cl-/Br- system in OA and RA.

This is an important point. Thank you again for the careful reading of our manuscript. These aspects were so far missing but are now comprehensively discussed in our revised manuscript version.

The authors further refer to a potential involvement of chlorine gas (Cl2) and dichlorine monoxide (Cl2O) in the abstract, however, this issue has been superficially addressed (lines 259-260) and thus does not justify any comment within the abstract.

Thank you again for the careful reading of our manuscript. The abstract was modified as suggested and the information about Cl2 and Cl2O deleted.

In addition to the requested changes, some smaller changes were made and some typos corrected. Our revised manuscript was now also seen by a native English speaker to improve the quality of the English style.

Round 2

Reviewer 1 Report

The authors have made some minor changes in the text. Their rebuttal and revision did not address previous concerns.This would leave this reviewer no choice other than refer the authors for previous concerns  marked as major concerns.

No comments as major issues remained unaddressed.

Author Response

The manuscript does not address recent development in the neutrophil field. This reviewer believes that references for other reviews on specific topics have limited value.

We feel sorry that you are not satisfied with the changes we have performed during the first revision. However, the focus of our review is on the chemical aspects of tissue destruction by MPO and/or HOCl and to a lesser extent on the medical and cellular aspects (cf. the title of our manuscript, please). We are neither physicians nor biologists. Therefore, the incorporation of some (nicely written) reviews is in our opinion a suitable way to address some complex issues and to save space in the journal.

It is unclear if this manuscript deals with RA or other types of arthritis. Clinical relevance of targeting MPO or HOCl should be explained ore explicitly.

It is (and actually was) explicitly stated that RA and osteoarthritis are of interest in our review. RA is characterized by a degradation of the components of the synovial fluid while osteoarthritis leads to the degeneration of the cartilage tissue. ROS contribute in both cases. Although the use of MPO inhibitors or scavengers of HOCl was already comprehensively discussed, this aspect is now even more highlighted. Thank you for this helpful comment.

The authors concept that more HOCl produces more damage would unlikely to explain the differences in clinical manifestation of different types of arthritis in spite of presence of neutrophils.

Many years ago, we studied the state of synovial fluid from patients suffering from rheumatoid arthritis by NMR. There was a clear correlation between the MPO activity and the content of degradation products [PMID: 8744012]. Therefore, we do not understand this critique. Nevertheless, we have emphasized this aspect in our revised manuscript version and emphasized that the medical aspects are not within the focus of the review.

Many novel developments in the neutrophil field have not been discussed in depth or in context with arthritis.

As far as we can say, this point was already addressed during the first revision step. Thus, we already added a chapter for NETs as you mentioned. As already mentioned above, we are no clinicians. Thus, we did not address the clinical outcome and/or guidelines to a significant extent but give rather some detailed references for the interested reader. If you have something helpful suggestions which kind of new aspects should be included and are helpful for experimental researchers, please give us at least a few keywords. Therefore, no changes in the manuscript were performed.

The authors provide a lengthy description of the well known chemistry of MPO and formation of HOCl, whereas clinical manifestations and neutrophil functional plasticity haven't been considered in depth in the text.

As already indicated twice (vide supra) we are not real experts in cell biology. Since a review should be written by experts in the field, our focus is on fields where we are most experienced. Surely not in the field of "neutrophil functional plasticity". Thank you for your kind understanding.

The authors have made some minor changes in the text. Their rebuttal and revision did not address previous concerns. This would leave this reviewer no choice other than refer the authors for previous concerns marked as major concerns.

We respectfully disagree with the reviewer. We have done much more than "minor changes"! This is obvious from the rebuttal letter and the marked changes during the first revision step. Thank you anyway for the help regarding the potential improvement of our manuscript.

Reviewer 3 Report

The authors have adequately responded to my previous comments raised. They have adequately covered NETs as well as addressed the role of HOCl to modify the plasmalogen moiety in rheumatic diseases to undersocre the role of myeloperoxidase and its major oxidant, HOCl, that has been confirmed in a series of other inflammatory diseases, and is presented here also in rheumatic diseases. The amended version is nicely supported by a series of figures to guide authors within the field of myeloperoxidase-mediated adverse effects during inflammation.

The authors have extensively covered the role of MPO and in particular the MPO-H2O2-Cl- system from activated neutrophils in rheumatic diseases and have further put major emphasis on its role in the progression of these diseases. Although other cell types beyond neutrophils may definitively play a role in these diseases, the primary focus of this review article is devoted to the eminent role of neutrophils and this has been nicely worked out

Author Response

Reviewer 3:

The authors have adequately responded to my previous comments raised. They have adequately covered NETs as well as addressed the role of HOCl to modify the plasmalogen moiety in rheumatic diseases to undersocre the role of myeloperoxidase and its major oxidant, HOCl, that has been confirmed in a series of other inflammatory diseases, and is presented here also in rheumatic diseases. The amended version is nicely supported by a series of figures to guide authors within the field of myeloperoxidase-mediated adverse effects during inflammation. The authors have extensively covered the role of MPO and in particular the MPO-H2O2-Cl- system from activated neutrophils in rheumatic diseases and have further put major emphasis on its role in the progression of these diseases. Although other cell types beyond neutrophils may definitively play a role in these diseases, the primary focus of this review article is devoted to the eminent role of neutrophils and this has been nicely worked out.

Thank you for your kind words. We are very happy that you are satisfied with the way we have dealt with your comments.